# Feasibility Demonstration of THz Wave Generation/Modulation Based on Photomixing Using a Single Wavelength-Tunable Laser

**Takashi Shiramizu [1], Naoya Seiki [1], Ryo Matsumoto [1], Naoto Masutomi [1], Yuya Mikami [1], Yuta Ueda [2] and Kazutoshi Kato [1,\***

[1]  Graduate School of Information Science and Electrical Engineering, Kyushu University, Fukuoka 819-0395, Japan
[2]  NTT Device Technology Laboratories, NTT Corporation, Atsugi 243-0198, Japan
\*    Correspondence: kato@ed.kyushu-u.ac.jp; Tel.: +81-92-802-3753

**Abstract:** The photomixing of two lightwaves is one of the promising methods of generating a terahertz (THz) wave. The conventional photomixing system consisting of two lasers and a modulator results in large transmitter volumes and high power consumption. To solve this issue, we devised a novel THz wave generation and modulation system based on photomixing using a single wavelength-tunable laser in combination with delayed self-multiplexing. We successfully demonstrated the feasibility of 300-GHz wave generation and modulation.

**Keywords:** terahertz wave; photomixing; UTC-PD; wavelength-tunable laser

## 1. Introduction

In recent years, the data traffic for wireless communications has been rapidly increasing [1] due to users' huge consumption of multimedia services. To deal with this issue, an electromagnetic wave with a frequency ranging from 300 GHz to 3 THz, which we call the THz wave, has attracted considerable attention due to its high potential to enhance the mobile communication band [2–8]. As for a THz wave modulation technique, there are two approaches: electronics-based and photonics-based methods. In the former method, the modulated THz waves are generated by upconverting an intermediate-frequency (IF) electrical signal to the THz-wave band [9–12]. Therefore, the modulation bandwidth is limited to the IF electrical signal bandwidth, while in the latter method, the modulated THz waves are generated by the photomixing of lightwaves that are modulated with an optical modulator [13,14]. Since the lightwave has a larger bandwidth than the IF electrical signal, the photomixing technique is more applicable for modulation with broadband signals compared to the electronics-based method. In addition, combining arrayed optical phase shifters and phased array antennas [15–18] in the photomixing technique enables beamforming and steering as well as other technologies such as using diffraction can [19]. For these reasons, the photomixing technique is one of the promising methods of THz wave generation and modulation for broadband wireless transmission.

In a conventional photomixing technique, one of the lightwaves from a laser is modulated by an optical modulator and coupled with a lightwave from another laser [20–22], as shown in Figure 1a. In another conventional method, two lightwaves from two lasers are coupled and then modulated [23], as shown in Figure 1b. Both methods require two lasers, driving circuits, temperature controllers, and an optical modulator. On the other hand, an optical frequency comb generator (OFCG) followed by a wavelength selective filter (WSF) makes two lightwaves as shown in Figure 1c [24–27]. Though the OFCG uses a single lightwave from a laser, it requires the WSF, which has complexity and a considerable volume.

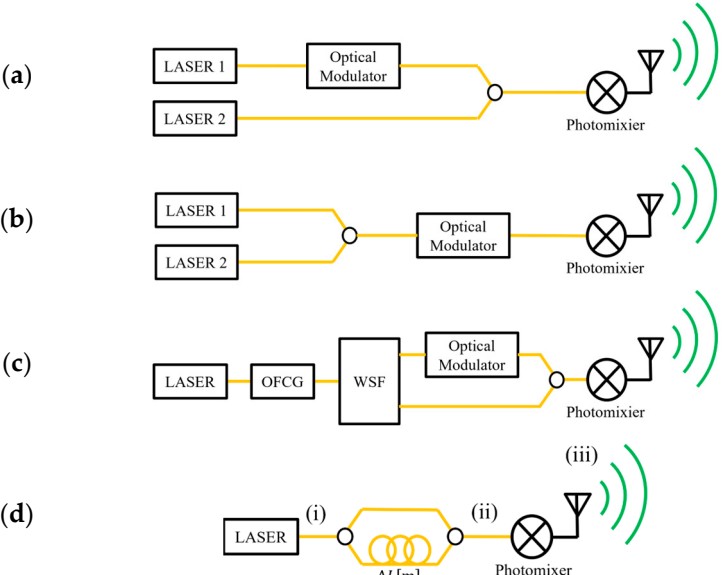

**Figure 1.** Diagrams of conventional terahertz wave generation and modulation systems based on photomixing technologies, (**a**) with two lasers and an optical modulator before an optical coupler, (**b**) with two lasers and an optical modulator after an optical coupler, and (**c**) with a single laser using an optical frequency comb generator. (**d**) Diagram of proposed terahertz wave generation and modulation system based on photomixing technologies with a single wavelength-tunable laser.

To deal with the above issues, we propose a novel photomixing system using only a single wavelength-tunable laser to generate and also modulate a THz wave. This system will contribute to low complexity and low power consumption of THz-wave transmitters for future broadband wireless communications.

## 2. Principle of THz Wave Generation and Modulation Using a Single Wavelength-Tunable laser

A system for generating a THz wave through conventional photomixing uses two lightwaves. The electric fields of these lightwaves, $E_1$ and $E_2$, are described as follows:

$$E_1 = A_1\exp\{j(2\pi f_1 t - k_1 z + \varphi_1)\}, \tag{1}$$

$$E_2 = A_2\exp\{j(2\pi f_2 t - k_2 z + \varphi_2)\} \tag{2}$$

where $A_1$ and $A_2$ are the amplitudes, $f_1$ and $f_2$ ($f_1 < f_2$) are the optical frequencies, $k_1$ and $k_2$ are the wave numbers, $\varphi_1$ and $\varphi_2$ are the initial phases, and z is an optical path length. The lightwaves are coupled by an optical coupler (OC) and inputted into a photomixer which converts the power of lightwaves into a photocurrent. The photocurrent $I$ is proportional to the square of the sum of the electric fields as follows [28].

$$I \propto |E_1 + E_2|^2 = A_1^2 + A_2^2 + 2A_1 A_2\cos\{2\pi(f_2 - f_1)t - (k_2 - k_1)z + (\varphi_2 - \varphi_1)\} \tag{3}$$

If the difference between $f_1$ and $f_2$ is equal to the frequency $f_{THz}$, which is in the THz-wave band ($f_2 - f_1 = f_{THz}$), an antenna integrated with the photomixer radiates a THz wave.

Figure 1d shows the configuration of a proposed system that generates and modulates a THz wave using only a single wavelength-tunable laser instead of a pair of lasers and a modulator. Firstly, the laser alternately switches its optical frequency between $f_1$ and $f_2$ to form segmented lightwaves where the frequency difference between $f_1$ and $f_2$ is $f_{THz}$, and the holding time at each frequency is $T_h$ as shown in Figure 2a. Secondly, the lightwave from the wavelength-tunable laser is split into two optical paths by an optical splitter (OS).

The lightwave at one of the two optical paths is delayed by an optical delay line (ODL) with a length of $\Delta L$ designed as follows.

$$\Delta L = \frac{cT_h}{n} \tag{4}$$

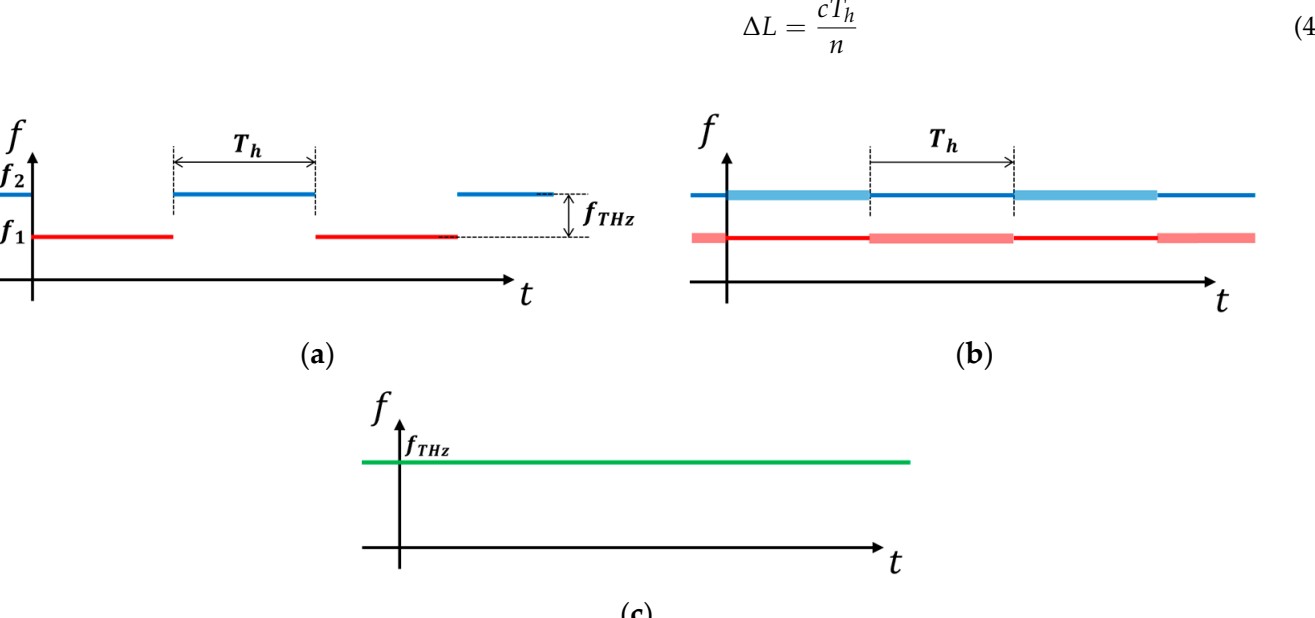

**Figure 2.** (**a**) The optical frequency of the lightwave from a wavelength-tunable laser at (i) of Figure 1d. (**b**) The optical frequency of the lightwave before photomixing at (ii) of Figure 1d. (**c**) The frequency of the terahertz wave generated by the photomixer at (iii) of Figure 1d.

That is the same as the length of each segmented lightwave at the optical path, where $c$ and $n$ are the speed of light and the effective refractive index of the optical fiber, respectively. Then, the lightwaves from two paths are coupled by an OC. We call this split-delayed-coupled operation 'delayed self-multiplexing.' As a result, two lightwaves of different frequencies always exist in an optical fiber as shown in Figure 2b. Finally, these two lightwaves are converted to a constant THz wave with a frequency of $f_{\text{THz}}$ by photomixing, as shown in Figure 2c. Here, a deviation of the delay length from $\Delta L$ results in discontinuity of the THz wave. While an alternating optical frequency change generates a constant THz wave, a static optical frequency does not result in a THz wave. In other words, a THz wave is generated only at a time when the optical frequency of the wavelength-tunable laser changes. Thus, the proposed system also generates a modulated THz wave described as follows. When the optical frequency of the lightwave is kept at $f_1$ or $f_2$ (Figure 3a,d), the lightwave inputted to the photomixer does not contain two frequencies (Figure 3e,h), and a THz wave is not generated (Figure 3i,l). On the contrary, when the optical frequency changes from $f_1$ to $f_2$ and vice versa (Figure 3b,c), two lightwaves with different optical frequencies (Figure 3f,g) are inputted into the photomixer and a THz wave is generated (Figure 3j,k). Thus, an optical frequency modulation encodes a differential on–off keying (OOK) modulation. For example, when the optical frequency is switched as $f_1$-$f_2$-$f_2$-$f_1$-$f_1$, the THz wave is modulated as 1–0–1–0.

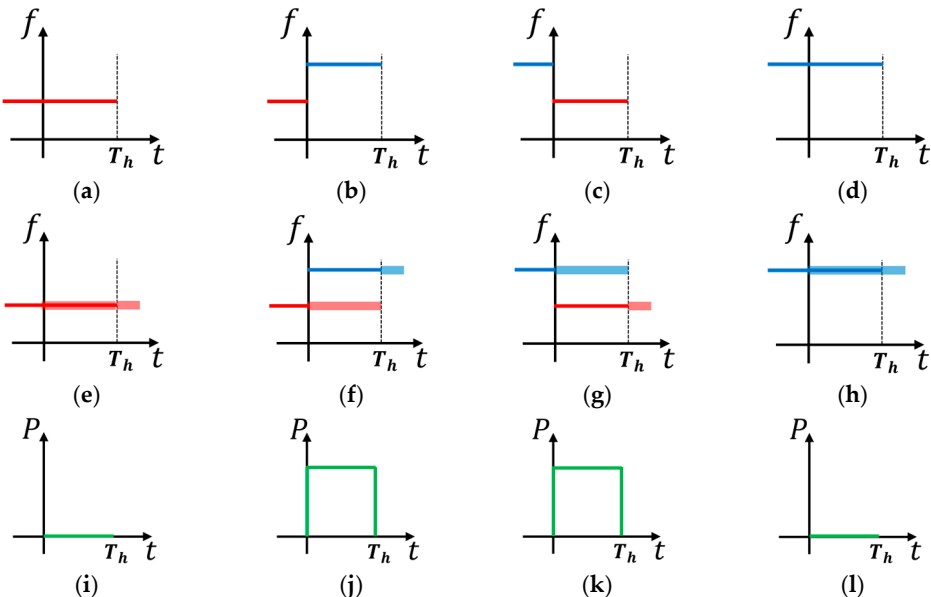

**Figure 3.** Optical frequency of the wavelength-tunable laser when it changes from (**a**) $f_1$ to $f_1$, (**b**) $f_1$ to $f_2$, (**c**) $f_2$ to $f_1$, (**d**) $f_2$ to $f_2$. Optical frequencies of lightwaves at the photomixer when the optical frequency changes from (**e**) $f_1$ to $f_1$, (**f**) $f_1$ to $f_2$, (**g**) $f_2$ to $f_2$, (**h**) $f_2$ to $f_2$. The intensity of terahertz wave when the optical frequency changes from (**i**) $f_1$ to $f_1$, (**j**) $f_1$ to $f_2$, (**k**) $f_2$ to $f_1$, (**l**) $f_2$ to $f_2$.

## 3. Experimental Setup

The experimental setup for the proposed THz wave generation and modulation using a single wavelength-tunable laser is shown in Figure 4. A pulse pattern generator (PPG) generated a voltage bit pattern at 150 Mbit/s (corresponding pulse width $T_h$ is 6.67 ns), and it was amplified by a power amplifier (PA). With a DC bias voltage through a bias tee, the voltage bit pattern was applied to the electrode of a wavelength-tunable laser. A reflection-type transversal filter (RTF) laser [29] was used in this experiment. The RTF laser can switch its optical frequency within 500 ps with a narrow linewidth (less than 350 kHz) [30]. The voltage bit pattern to the RTF laser causes switching of the optical frequency between 193.230 THz and 193.528 THz whose difference is approximately 300 GHz. Then, the lightwave was amplified through an erbium-doped fiber amplifier (EDFA) and introduced to the delayed self-multiplexer in which the lightwave was split by the OS and one of them was delayed by an ODL. The delayed length $\Delta L$ was set at approximately 135 cm according to Equation (4). After that, the lightwaves coupled with an OC were inputted into a photomixer module. We used a uni-traveling-carrier photodiode (UTC-PD) [31] as a photomixer. The radiated THz wave from the photomixer module was propagated in a waveguide and detected by a Schottky barrier diode (SBD) integrated with a WR2.8 waveguide whose passband is from 260 GHz to 400 GHz. The output voltage of the SBD was then amplified by a low-noise amplifier (LNA) and its waveform was observed by an oscilloscope (OSC).

First, we applied the 16 bits voltage pattern of "0101010111111111" to the RTF laser, where 1 and 0 correspond to optical frequencies of 193.53 THz and 193.23 THz, respectively. It was expected that the first 8 bits would generate a constant THz wave and the following 8 bits would not, as shown in Figure 5a. Next, we applied a 16 bits pseudo-random pattern of "1001111000010101", to the RTF laser to testify the feasibility of the differential OOK modulation that is expected to generate a THz wave with a pattern of "0101000100011111" as shown in Figure 5b.

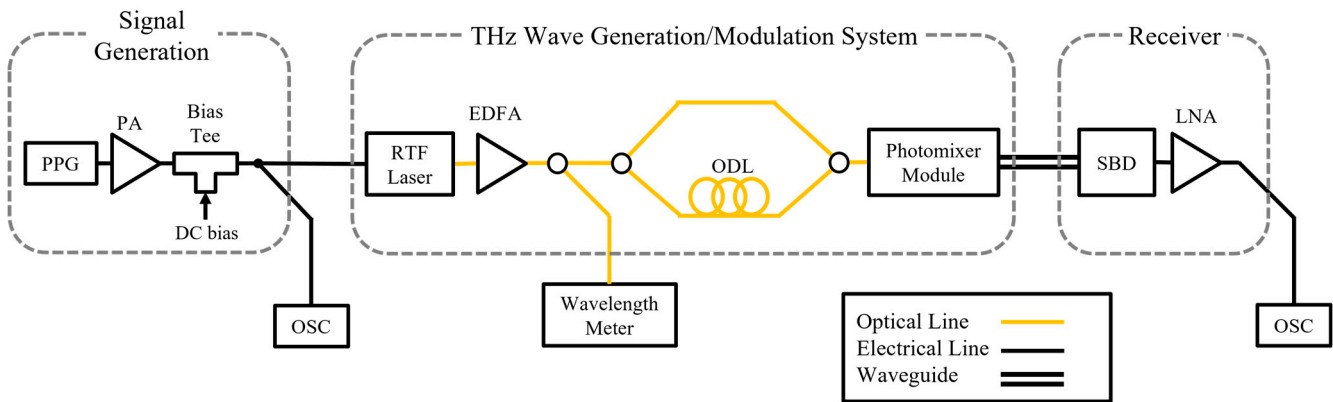

**Figure 4.** Experimental setup for photomixing using a single RTF laser.

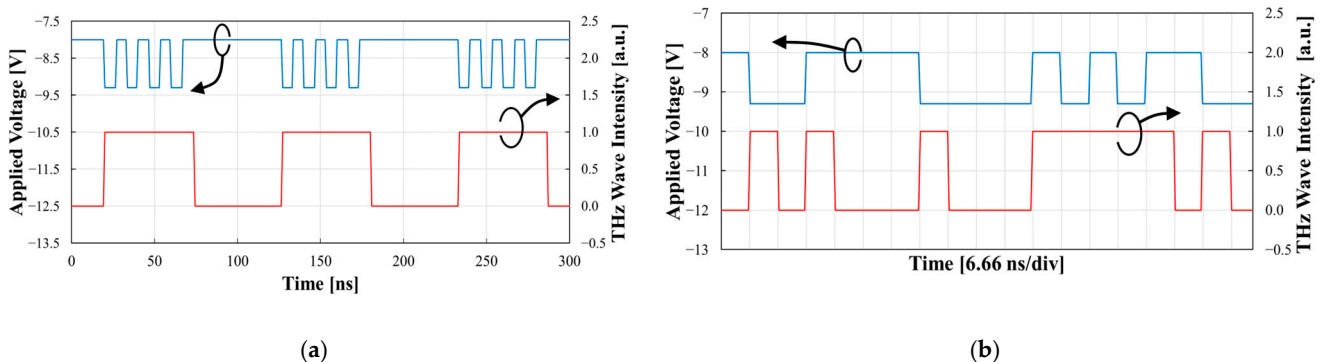

(**a**)                                                                              (**b**)

**Figure 5.** Designed waveform of applied voltage (upper) and expected terahertz-wave intensity (lower). (**a**) at the pattern of "0101010111111111"; (**b**) at the pattern of "1001111000010101".

## 4. Experimental Result and Discussion

Figure 6a shows the voltage pattern "0101010111111111" applied into the RTF laser (upper) and the THz-wave intensity obtained by the SBD (lower). During the experiment, the optical frequencies emitted from the RTF laser were measured to be 193.230 THz and 193.528 THz. The results show that a 300 GHz THz wave was generated at the first 8 bits pattern (01010101), and was not at the following 8 bits pattern (11111111) as expected in Figure 5a. Therefore, this shows the successful THz wave generation with a single wavelength-tunable laser. The unexpected dips on the waveform of the THz wave originated from a deviation of the delayed length from an ideal length calculated by Equation (4) and a switching response of the RTF laser.

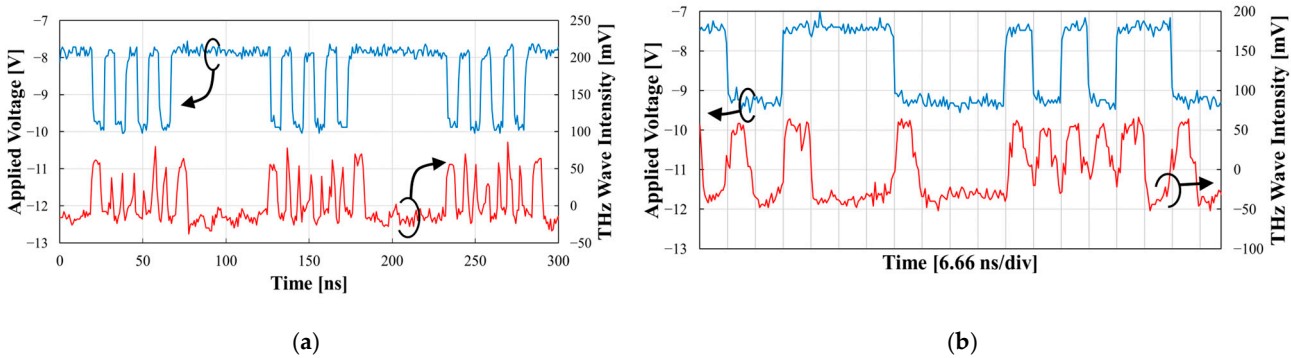

(**a**)                                                                              (**b**)

**Figure 6.** Measured waveform of applied voltage (upper) and the terahertz-wave intensity (lower). (**a**) at the pattern of "0101010111111111"; (**b**) at the pattern of "1001111000010101".

The former would be decreased by adjusting $\Delta L$ at the length exactly corresponding to $T_h$. The latter would be improved by introducing a pre-emphasized voltage waveform such as overshoot-based feed-forward control so as to utilize the short response time of the RTF laser within 500 ps [29]. As a feasibility demonstration of a modulated THz wave, the voltage bit pattern "1001111000010101" shown by the upper curve in Figure 6b was applied to the RTF laser. The observed THz-wave intensity shown by the lower curve is in good agreement with the expected differential OOK waveform (Figure 5b). The dips also appear in the pattern and would be decreased in the same manner described in the case of a constant THz wave. Otherwise, even with the dips, this modulation technique can be utilized as a conversion from an optical frequency modulation to a differential-RZ-THz-wave signal. As a future study, the bit error rate of the modulated data could be measured by using a decoder of the differential OOK signal. In addition, Gbit/s-class data rate can be achieved with this system by shortening wavelength switching time down to less than a nanosecond as well as by using multi-level frequency shift keying (FSK) modulation.

## 5. Conclusions

We devised a novel THz wave generation and modulation system based on photomixing that uses only a single wavelength-tunable laser for low complexity and low power consumption of THz wave transmitters for future wireless communications. The THz wave can be generated by switching an optical frequency of a wavelength-tunable laser in combination with delayed self-multiplexing. Using the RTF laser, we successfully demonstrated the feasibility of the THz wave generation at 300 GHz, as well as its OOK modulation with a 150 Mbit/s pulse pattern.

**Author Contributions:** Conceptualization, T.S. and K.K.; methodology, T.S., N.S., R.M., N.M., Y.M., Y.U. and K.K.; validation, T.S., N.M., Y.M. and K.K.; investigation, T.S., N.M., Y.M. and K.K.; resources, Y.U. and K.K.; data curation, T.S.; writing—original draft preparation, T.S.; writing—review and editing, Y.M. and K.K.; visualization, T.S.; supervision, Y.M. and K.K.; project administration, K.K.; funding acquisition, K.K. All authors have read and agreed to the published version of the manuscript.

**Funding:** This work was supported in part by the commissioned research by National Institute of Information and Communications Technology (NICT) #02801, #00901, the MIC/SCOPE #195010002, and JSPS KAKENHI Grant Number: JP20H00253, JP21K18730.

**Institutional Review Board Statement:** Not applicable.

**Informed Consent Statement:** Not applicable.

**Data Availability Statement:** Data available on request.

**Conflicts of Interest:** The authors declare no conflict of interest.

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
