# Peer review of "Feasibility Demonstration of THz Wave Generation/Modulation Based on Photomixing Using a Single Wavelength-Tunable Laser"

_photonics, doi:10.3390/photonics10040369_

Round 1

Reviewer 1 Report

The manuscript by Takashi Shiramizu, Naoya Seiki, Ryo Matsumoto, Naoto Masutomi, Yuya Mikami, Yuta Ueda and Kazutoshi Kato includes a systematic, interesting and very valuable proof-of-principle study concerning a novel THz wave generation and modulation technique using only one tunable laser in the photomixing methodology. The authors give a very valuable and good overview about the state-of-the-art, propose the concept, explain the methodology of the generation, establish a corresponding set-up and demonstrate it via an experimental characterization. The paper is of very high value for the community. I strongly recommend publication befor some minor points are modified.

 -          Througout the text and the figure captions, the authors use terahertz and only in the title and the headings they use the abbreviation THz. I recommend to use the abbreviation  everywhere, after having it introduced in the abtract (e.g.).

-          I recommend a rephrasing in line 12:  modulator results in large transmitter volumes and high power consumption.

 -          The authors mention the important THz beamsteering in line 33. I recommend to mention and cite here also an alternative published ten years ago by Y. Monnai, et al: Terahertz beam focusing based and variable focusing using programmable diffraction gratings. Optics Express Vol. 21, pp. 2347-54 (2013).

Reviewer 2 Report

In this manuscript, the authors propose and experimentally demonstrate a new method to generate and modulate terahertz waves with the use of a single wavelength-tunable laser and a delayed self-multiplexing technique. Compared to conventional systems consisting of two lasers and an optical modulator, the authors’ system can provide a lower complexity and lower power consumption. They first theoretically illustrate the working principle of terahertz generation with delayed self-multiplexing. The delayed self-multiplexing can also be used as an optical frequency modulation, which encodes a differential on-off-keying (OOK) modulation. They use the experiment of two designed voltage patterns, which control the wavelength of the tunable laser, to demonstrate the successful generation and modulation of terahertz waves at 300 GHz.

There are a number of issues or questions that I believe should be addressed before publication.

Major comments:

1) In Fig. 6(a) and (b), there are fluctuations in the generated terahertz waveform (“dips” described by the authors). The authors explained the fluctuations by two reasons. The first is that they are originated from a deviation of the delayed length from an ideal length calculated by Eq. (4), and the authors suggest that they can be eliminated down to almost zero by adjusting ΔL at the length exactly corresponding to Th. This can be verified in the same setup by changing Th in the pulse pattern generator with a fixed length of optical delay line. It would also increase the robustness of the paper if they can show how the detuning of Th from the ideal Th affects the terahertz wave generation.

2) Following the previous point, the authors’ second suggestions to mitigate the fluctuation issue in the terahertz waveform is to introduce a “pre-emphasized voltage waveform so as to utilize the short response time of the RTF laser within 500 ps [28]”. It is not clear to me what pre-emphasized voltage waveform means. Does it mean a better-selected voltage range, a shorter rising/falling time, or a pre-defined voltage pattern? The authors should clarify more in the paper.  

3) The main application of the authors’ terahertz wave generation and modulation system is for future wireless communication. However, the demonstrated modulation speed (150 Mbit/s) is still far from the high data rate promised by terahertz communication (e.g., Ding, J. et al. THz-over-fiber transmission with a net rate of 5.12 Tbps in an 80 channel WDM system. Opt. Lett. 47, 3103 (2022)). It would be beneficial to discuss the possibility in the authors’ system to achieve higher-data-rate communication.

Minor but still important comments:

1) The demonstrated differential on-off-keying modulation is not widely used in the state-of-the-art communication system. A discussion of the compatibility of more advanced modulation method to the terahertz system would strengthen this paper.

2) I suggest moving Fig. 2(a) to Fig. 1 or combining Fig. 1 and Fig. 2. Since one of the main objective of this work is to demonstrate a more compact and simpler terahertz wave generation and modulation system, it would be good to compare the block diagrams of different systems in the same figure.

3) In Equation (3), I believe there is missing coefficient of “2” in front of “A1A2” in the last term.

Typos:

1) In line 117, the number should be 4.

2) In line 139, the number should be 5.

Round 2

Reviewer 2 Report

Thanks for the revised manuscript. It addresses the questions I have in the previous report. I recommend its publication.